# The Expression of Hemagglutinin by a Recombinant Newcastle Disease Virus Causes Structural Changes and Alters Innate Immune Sensing

**DOI:** 10.3390/vaccines9070758

**Published:** 2021-07-07

**Authors:** Fiona Ingrao, Victoria Duchatel, Isabel Fernandez Rodil, Mieke Steensels, Eveline Verleysen, Jan Mast, Bénédicte Lambrecht

**Affiliations:** 1Service of Avian Virology and Immunology, Sciensano, 1180 Brussels, Belgium; victoria.duchatel@sciensano.be (V.D.); mieke.steensels@sciensano.be (M.S.); benedicte.lambrecht@sciensano.be (B.L.); 2Institut de Biologie et Médecine Moléculaire, Université Libre de Bruxelles, 6041 Gosselies, Belgium; isabelherge@gmail.com; 3Service of Trace Elements and Nanomaterials, Sciensano, 1180 Brussels, Belgium; eveline.verleysen@sciensano.be (E.V.); jan.mast@sciensano.be (J.M.)

**Keywords:** Newcastle disease virus, avian influenza, vector vaccines, glycoproteins, innate immunity

## Abstract

Recombinant Newcastle disease viruses (rNDV) have been used as bivalent vectors for vaccination against multiple economically important avian pathogens. NDV-vectored vaccines expressing the immunogenic H5 hemagglutinin (rNDV-H5) are considered attractive candidates to protect poultry from both highly pathogenic avian influenza (HPAI) and Newcastle disease (ND). However, the impact of the insertion of a recombinant protein, such as H5, on the biological characteristics of the parental NDV strain has been little investigated to date. The present study compared a rNDV-H5 vaccine and its parental NDV LaSota strain in terms of their structural and functional characteristics, as well as their recognition by the innate immune sensors. Structural analysis of the rNDV-H5 demonstrated a decreased number of fusion (F) and a higher number of hemagglutinin-neuraminidase (HN) glycoproteins compared to NDV LaSota. These structural differences were accompanied by increased hemagglutinating and neuraminidase activities of rNDV-H5. During in vitro rNDV-H5 infection, increased mRNA expression of TLR3, TLR7, MDA5, and LGP2 was observed, suggesting that the recombinant virus is recognized differently by sensors of innate immunity when compared with the parental NDV LaSota. Given the growing interest in using NDV as a vector against human and animal diseases, these data highlight the importance of thoroughly understanding the recombinant vaccines’ structural organization, functional characteristics, and elicited immune responses.

## 1. Introduction

Newcastle disease (ND) and highly pathogenic avian influenza (HPAI) are two highly contagious and economically devastating notifiable poultry diseases [1,2]. The continuous threat they represent to the poultry sector worldwide emphasizes the need for high-level biosecurity measures, strong surveillance strategies, and efficient vaccination programs [3,4,5]. Both attenuated and inactivated vaccines have been extensively and successfully used to protect poultry from infectious diseases, but they can also have drawbacks, such as the interference of maternal antibodies, that have been previously reviewed [6,7]. The development of new vaccination strategies is an answer to the call for more protective vaccines that would overcome the challenges faced by classical immunization. The use of recombinant viral vector-based vaccines expressing one or several foreign genes represents a promising vaccination strategy. Live recombinant vaccines have the advantage of eliciting cellular and mucosal responses as well as humoral immunity. Herpesvirus of turkey, NDV LaSota, and fowlpox viruses are widely used veterinary vaccine virus backbones [8] and NDV is also considered as a promising viral vector vaccine candidate against human diseases such as HPAI or the severe acute respiratory syndrome-associated coronaviruses [9]. The protection afforded by these vaccines is validated by testing their potency following an infection by a virulent pathogen related to the viral vector or the foreign gene expressed, and by identifying the host immune responses elicited. However, a shortcoming of this approach is the lack of systematic investigation of the impact of foreign gene insertion into the genome of the vector on its structure and biological functions. Recombinant NDV vectored vaccines expressing the protective antigen H5 (rNDV-H5) have demonstrated their efficacy in protecting specific pathogen-free (SPF) chickens against both homologous and heterologous HPAI H5 and velogenic ND challenges [10,11,12,13,14]. NDV infection is initiated by the attachment of the virus through the binding of hemagglutinin-neuraminidase (HN) glycoprotein to the sialic-acid receptor at the surface of the host cell [15]. Like other viruses included in the *Paramyxoviridae* family, NDV is known to enter its target cell through direct fusion with the cell membrane, and it has been suggested to use a caveolae-dependent endocytic pathway as an alternative route for viral entry [16,17]. A previous study examining a recombinant NDV expressing the glycoprotein GP of the Ebola virus showed that it used GP-dependent macropinocytosis as a major cell entry pathway, indicating that the foreign GP can function as an entry protein [18]. The AIV entry process begins with the binding of the hemagglutinin (HA) to sialic acids at the cell surface and the internalization of the viral particle by endocytosis. The low pH within the endosome triggers conformational changes in the HA, exposing the fusion peptide and inducing the fusion between the virus and the endosomal membrane [19]. The ability of rNDV-H5 to use an H5-dependent entry pathway under certain conditions, such as the presence of ND maternal antibodies, has been suggested by a previous study [20]. The use of this alternative entry pathway could, therefore, affect vaccine-induced immune responses. Because the latter may differ from the immune responses induced by the parental NDV LaSota, their characterization would improve the understanding of protection outcomes previously observed with rNDV-H5 immunization. In this study, the analyses focused on innate responses known to be involved in the regulation and orientation of subsequent adaptive responses [21]. Pathogen recognition by innate immune system is mediated through the sensing by pattern recognition receptors (PRRs). The activation of these receptors generates signals that trigger intracellular cascades resulting in the production of key soluble mediators that influence the polarization of adaptive immune responses [22]. Among PRRs, toll-like receptors (TLRs) -3 (TLR3) and -7 (TLR7) are important virus sensors capable of recognizing nucleic acids in intracellular compartments such as endosomes. Melanoma differentiation-associated gene 5 (MDA5) and laboratory of genetics and physiology 2 (LGP2) are RNA-sensing PRRs expressed in the cytoplasm that play a key role in the activation of the viral sensing pathway [23,24]. The recognition of nucleic acids derived from pathogens during an infection ultimately leads to the production of type-I interferons (IFNs) that mediate the antiviral response [25] and cytokines that influence the polarization of adaptive immune responses [26,27].

To determine if the recombinant NDV-H5 retains the structural and functional characteristics of the parental NDV LaSota strain, the present study compared the structural organization and enzymatic activity of surface glycoproteins, and the recognition of both viruses by innate immune sensors.

## 2. Materials and Methods

### 2.1. Chickens

SPF White Leghorn chickens were hatched from embryonated eggs purchased from Lohmann Valo (Cuxhaven, Germany). After hatching, the chickens were housed in biosecurity level 3 isolators. Feed and water were provided ad libitum throughout the experimental period.

### 2.2. Vaccines and Viruses

The rNDV-H5 vaccine expressing a modified H5 ectodomain of human HPAI H5N1 clade 1 A/Vietnam/1203/04, and the NDV LaSota were provided by Lohmann Animal Health GmbH (Germany) [12]. The H5 insert of the rNDV-H5 were modified into a low-pathogenic version to ensure vaccine safety, and the H5 transmembrane domain and cytoplasmic tail were replaced by those of the NDV F glycoprotein to allow surface expression [12,28]. The H5 gene was inserted between the phosphoprotein and matrix genes of the NDV genome, as it has been identified as the optimal insertion site for foreign gene expression [29]. The strains used in this study were amplified by inoculation into the allantoic cavity of 9–11 day-old embryonated specific pathogen free (SPF) eggs. Five days after inoculation or at the death of the embryo, allantoic fluids were harvested and the isolates were titrated on primary chicken embryo fibroblasts (CEFs) to determine the tissue culture infectious dose (TCID50/mL) [30,31]. For immunogold electron microscopy analyses, viral strains were purified by differential centrifugation on a sucrose gradient, as previously described [32].

### 2.3. Cells and Monoclonal Antibodies

CEFs were cultured in complete medium composed of a mixture of Leibovitz’s L15 and McCoy’s 5A (1:1) media (Gibco, Thermo Fisher Scientific, Waltham, MA, USA) supplemented with 2 % heat-inactivated fetal calf serum, 2 mM L-Glutamin, and 50 µg/mL gentamycin (Gibco, Thermo Fisher Scientific, Waltham, MA, USA) at 37 °C under 5% CO_2_.

NDV F and HN glycoproteins and AIV H5 were detected using previously described in-house monoclonal antibodies (mAbs): mouse anti-NDV F 1C3 (IgG1), mouse anti-NDV HN 4D6 (IgG2a), and mouse anti-AIV H5 5A1 (IgG1) [33,34].

### 2.4. Immunogold Electron Microscopy

Glycoprotein expression on the rNDV-H5 and NDV LaSota surface was evaluated by the previously described immunogold labeling method [20] with minor modifications. Briefly, pioloform carbon-coated copper grids (Agar Scientific, Stansted Essex, UK) were pretreated with Alcian blue 8G (Gurr Microscopy Materials, Poole, UK) solution at 1 % *v/v* in water for 10 min at room temperature (RT). The rNDV-H5 and NDV LaSota were diluted in PBS to a final concentration of 75 µg/mL and adsorbed onto pretreated grids for 10 min at RT. Anti-F, anti-HN, and anti-H5 mAbs at 1:50 dilution in PBS supplemented with 2% of goat serum were then adsorbed to the grid. The number of gold particles at the surface of the virions was assessed in 50 representative virions. Images of immunogold-labeled virions were acquired on a Tecnai G2 Spirit electron microscope (FEI, Eindhoven, The Netherlands) using bright-field transmission electron microscopy mode. To take into account the pleomorphism of NDV viruses [35], the surface of each virion was measured and the number of gold labels was then expressed per 55,000 nm^2^ as an estimate of the number of gold particles per virion (#gold/virion). The mean surface of the virions was determined as 55,000 nm^2^.

### 2.5. Virus Neutralization

CEFs were seeded at a concentration of 5 × 10^5^ cells/mL in 96-well plates and incubated at 37 °C for 24 h. Two-fold serial dilutions of an initial concentration of 5 µg/mL of the mAbs were incubated with NDV LaSota or rNDV-H5 for 3 h at 37 °C in complete medium supplemented with 50 ng/mL L-1-tosylamido-2-phenylethylchloromethyl ketone (TPCK)-treated trypsin (Sigma Aldrich, St. Louis, MO, USA). Subsequently, CEFs monolayers were cultured with mixtures of mAbs and NDV LaSota or rNDV-H5, corresponding to a multiplicity of infection (MOI) of 0.01. After 24 h, half of the culture medium was replaced by fresh complete medium supplemented with TPCK-trypsin. CEFs were monitored daily over a 7 day period for the presence of a cytopathic effect.

### 2.6. Evaluation of Hemagglutinating and Neuraminidase Activities

The hemagglutinating activity of NDV LaSota and rNDV-H5 was evaluated by the standard hemagglutination assay [36]. Both viruses were serially two-fold diluted in triplicate from a starting titer of 10^7^ TCID50/mL and the HA titers were determined based on the lowest virus dilution at which full hemagglutination was observed.

The neuraminidase activity of NDV LaSota and rNDV-H5 HN was determined using the NA-fluor Influenza Neuraminidase Assay kit (Applied Biosystems, CA, USA) according to the manufacturer’s recommendations. Triplicates of two-fold serial dilutions of NDV LaSota and rNDV-H5 starting at a titer of 10^7^ TCID50/mL were analyzed and the neuraminidase activity was expressed as Relative Fluorescent Unit (RFU).

### 2.7. Immunofluorescence

Immunofluorescence was performed as previously described [37]. Briefly, CEFs cultured in 6-well plates and the monolayer was infected with either NDV LaSota or rNDV-H5 at an MOI of 1 and incubated at 37 °C for 1 h. The medium was then replaced by fresh complete medium without antibiotics and the CEFs were incubated at 37 °C for 0, 2, 6, 10, and 24 h. NDV F protein was labeled with 1:100 1C3 mAb, followed by 1:100 FITC-conjugated sheep anti-mouse IgG as secondary antibody (F6257, Sigma Aldrich, St. Louis, MO, USA). Fluorescence was detected using a Leitz SMLUX microscope with a Leica DFC420C camera and images were analyzed with the Leica Application Suite LAS V.4 program.

### 2.8. Tracheal Organ Cultures (TOCs) Infection

Tracheas were aseptically collected from nine 12-day-old SPF chickens and washed with warm complete culture medium containing DMEM (Gibco, Thermo Fisher Scientific, Waltham, MA, USA) supplemented with 100 U/mL penicillin (Kela Pharma, Sint-Niklaas, Belgium) and 1 mg/mL streptomycin (Sigma Aldrich, St. Louis, MO, USA). The upper part of the tracheas was dissected into 2–3 mm rings. The rings from the nine chickens were divided into three groups and cultured in pools of three per well in 1 mL of complete medium in a 12-well plate. Rings were cultured for 48 h at 39 °C in 5% CO_2_ atmosphere. The culture medium was then removed and replaced with 0.5 mL of viral inoculum at the titer of 10^6^ TCID50/mL in complete culture medium. Virus adsorption was carried out for 1 h at 39 °C, after which 1.5 mL of complete medium was added. The rings were collected after 0, 2, 6, 10 and 24 h post-infection (hpi) and were stored in pools of three in 200 µL in RNAlater solution (Applied Biosystems, Lennik, Belgium) at −80 °C until RNA extraction.

### 2.9. CEFs Infection

The CEFs were cultured and infected according to the protocol described above for immunofluorescence assay. At 0, 2, 6, 10, and 24 hpi, the medium was discarded and CEFs were detached using a solution of 0.25% Trypsin-EDTA (Thermo Fisher Scientific, Waltham, MA, USA). The infected CEFs of two wells were pooled and were later stored in RNA at –80 °C until analysis.

### 2.10. RNA Extraction and Real-Time Reverse Transcription (RT)-PCR

The RNA from infected TOCs and CEFs samples was extracted using the MagMAX-96 Total RNA Isolation kit (AM1830, Ambion, Applied Biosystems, Carlsbad, CA, USA). Synthesis of cDNA was performed using 250 ng of purified RNA using oligo(dT)_15_ primers (GoScriptTM Reverse Transcription System, A5001, Promega, Madison, WI, USA), according to the manufacturer’s instructions. The cDNA products were stored at -20 °C until further use. The relative expression of TLR3, TLR7, MDA5, [38], LGP2 [39], IFNα [40], and IFNβ [41] was measured by RT-PCR, according to a previously published protocol [42]. HMBS and RPLP0 [43] were selected as reference genes for normalization of RT-PCR results using the algorithm GeNorm (Biogazelle, Zwijnbeke, Belgium). Normalized gene expression was quantified as the fold change relative to the uninfected cells at time point 0 hpi according to the 2^−ΔΔCT^ method [44].

### 2.11. Statistical Analyses

Statistical analyses were performed using R statistical software and the results were visualized using the ggplot2 package for R [45]. Immunogold electron microscopy results were analyzed by Mann–Whitney–Wilcoxon test or Student’s paired *t*-test using permutation and, for normally distributed values, with one-way analysis of variance (ANOVA). The neuraminidase activity data were analyzed with one-way analysis of variance (ANOVA). Flow cytometry data were analyzed by ANOVA test using permutation or by Student’s paired *t*-test using permutation depending on the normality and homoscedasticity. RT-PCR data were analyzed with ANOVA and the non-parametric Kruskal–Wallis test, while innate immunity results on TOCs were analyzed with ANOVA only. *p*-values < 0.05 were considered statistically significant.

## 3. Results

### 3.1. The Recombinant Virus Expresses Higher Levels of HN Glycoproteins and Lower Levels of F on Its Surface Than the Parental NDV

Immunogold electron microscopy was conducted to evaluate the expression of F and HN at the surface of rNDV-H5 and to compare it to the distribution of these glycoproteins on parental NDV LaSota. Quantitative analysis of the labeling densities of F glycoprotein demonstrated a significantly lower expression at the surface of rNDV-H5 (4.3 ± 0.3 gold/virion) when compared to NDV LaSota (8.4 ± 0.5 gold particles/virion). In contrast, rNDV-H5 displayed a significantly higher number of HN molecules at its surface (25.8 ± 1.4 gold particles/virion) than NDV LaSota (18.4 ± 1.9 gold particles/virion) (Figure 1a). Immunogold labeling also confirmed the presence of H5 at the surface of rNDV-H5 (4.6 ± 0.7 gold particles/virion), while a background level of H5 labeling of 0.8 ± 0.2 gold particles/virion was detected on NDV LaSota’s surface.

The capacity to block the viral entry of NDV LaSota and rNDV-H5 using anti-F and HN monoclonal antibodies was evaluated using a neutralization test. The neutralization curves obtained using the anti-F monoclonal antibody were similar for both viruses (Figure 1b, left panel), while NDV LaSota was neutralized at 83% by anti-HN monoclonal antibody at the concentration of 0.63 µg/mL, which was significantly higher than the 17% of neutralization of rNDV-H5 at the same mAb concentration (Figure 1b, right panel).

### 3.2. rNDV-H5 Has Higher Hemagglutinating and Neuraminidase Activities Than Parental NDV LaSota

The hemagglutination assay demonstrated that rNDV-H5 retains the ability to agglutinate chicken erythrocytes (Figure 2a). Hemagglutinating activity of two-fold serially diluted NDV LaSota and rNDV-H5 at the initial concentration of 10^7^ TCID50/mL was last fully detected at the titer of 6.25 × 10^5^ TCID50/mL (1:8 dilution) and 7.8 × 10^4^ TCID50/mL (1:32 dilution), respectively.

The comparison of the neuraminidase activity showed that enzymatic activity of rNDV-H5 was significantly increased relative to that of the parental NDV LaSota at all virus titers tested (Figure 2b), which is in accordance with the higher HN content of rNDV-H5.

### 3.3. Innate Sensing of rNDV-H5 Is Mediated by TLR3, MDA5, and LGP2

CEFs were infected in triplicate with either NDV LaSota or rNDV-H5 at an MOI of 1. Representative images are shown in Figure 3a. CEFs infected with NDV LaSota and rNDV-H5 displayed a similar pattern of immunofluorescence for the F glycoprotein. However, infection kinetics differed slightly between both viruses, as NDV LaSota infection was more notable at 10 hpi than rNDV-H5.

Early immune responses induced following the infection of CEFs (Figure 3b) and TOCs (Figure 4) with rNDV-H5 and NDV LaSota were investigated through the evaluation of changes in the expression of genes associated with innate immune responses. The innate sensing of rNDV-H5 and NDV LaSota was first evaluated through the investigation of changes in the expression of PRRs TLR3, TLR7, MDA5, and LGP2.

A significantly increased expression of TLR3 at 2, 6, 10, and 24 hpi was observed in rNDV-H5 infections of CEFs, as compared with NDV LaSota. Expression of TLR7 was not significantly altered, regardless of the condition of infection. An increase in the expression of MDA5 and LGP2 was detected in CEFs infected with rNDV-H5 at 6, 10, and 24 hpi, as compared with NDV LaSota-infected cells.

The expression of TLR3 was significantly increased in the TOCs infected with rNDV-H5 compared to those infected with NDV LaSota at 6 hpi. No differences were observed in TLR7 gene expression between rNDV-H5- and NDV LaSota-infected TOCs. At 2 hpi, rNDV-H5-infected TOCs presented higher MDA5 gene expression than did those infected with NDV LaSota, and this trend was maintained at 6 and 10 hpi but was not significant. However, MDA5 expression was significantly higher in TOCs infected with NDV LaSota at 24 hpi compared to rNDV-H5 infection. The expression of LGP2 by rNDV-H5-infected TOCs was moderately upregulated at 2 and 6 hpi.

The expression of antiviral type-I IFNs, IFNα and IFNβ, was also evaluated to determine whether it correlated with changes in PRR expression. A relatively modest decrease in IFNα expression was observed in cells infected with rNDV-H5 compared to NDV LaSota CEFs infection. In contrast, expression levels of IFNβ were increased in CEFs infected with rNDV-H5 at 6, 10, and 24 hpi, compared with NDV LaSota infection. These changes mirrored the decreased expression levels of MDA5 and LGP2 at the same time point. None of the TOCs infections significantly affected the expression of antiviral IFNα and IFNβ, regardless of the tested time point.

## 4. Discussion

The structure of rNDV-H5 and the expression of H5 on the vaccine surface have previously been observed by immunogold electron microscopy [20]. The present study confirmed these results and found significant structural differences between rNDV-H5 and the parental LaSota NDV. Because the H5 gene is inserted upstream of the 3’ end of the genome of the rNDV-H5 construct, between phosphoprotein and matrix genes, the level of expression of downstream F and HN could be impacted and reduced [46,47]. However, the investigation of both surface glycoproteins distribution demonstrated a higher expression of HN and a lower expression of F by rNDV-H5 when compared to parental NDV LaSota, although HN is positioned more distally than F in the NDV genome. Earlier studies have shown that HN interacts with F and promotes its fusion activity [48,49]. During budding and virion assembly processes, HN and F are anchored into the viral envelope by the interaction of their cytoplasmic tail with the M protein [50]. The HA at the surface of rNDV-H5 is expressed as a chimeric protein whose H5 transmembrane domain and cytoplasmic tail have been replaced by those of the NDV F glycoprotein. Nayak et al. (2009) demonstrated that chimeric H5 was incorporated more efficiently into the NDV when compared to unmodified H5, but reduced replication of the recombinant vector without impacting its pathogenicity, suggesting that H5 incorporation may have affected the NDV assembly process [13]. The slight delay in the infection kinetics of CEFs with rNDV-H5 and the lower level of expression of F protein observed in the present study suggest that the incorporation of the chimeric HA might be at the expense of F incorporation. The structural impact of H5 incorporation could be confirmed by comparing the expression levels of F protein on the surface of recombinant NDV expressing chimeric or native HA.

Balanced hemagglutinin and neuraminidase enzyme activities are known to be critical for the outcome of an infection [48,51,52]. Following viral budding, the newly synthesized virus particles bound to the sialic acids of the host cells are released by the sialidase activity of the neuraminidase (reviewed in [53]). A weak neuraminidase activity was found to be counterbalanced by a decreased HA affinity, maintaining the ability of the virus to replicate efficiently [54,55]. The results of the present study suggest that the presence of H5 on the surface confers a slightly higher hemagglutinating activity to rNDV-H5 compared to the parental LaSota NDV. In addition to the increased neuraminidase activity carried by rNDV-H5, these results raised the hypothesis that the presence of H5 presumably disrupted the hemagglutinin-neuraminidase balance and may have been compensated by an enhanced neuraminidase activity as a result of a higher expression of HN protein. The comparison of the HN sequences of rNDV-H5 and NDV LaSota should be performed to evaluate the hypothesis that the increase in neuraminidase activity is indeed correlated to the increased number of HN proteins on the surface of the recombinant virus and not due to acquired compensatory mutations, as previously demonstrated for AI in the case of an imbalance of hemagglutinating and neuraminidase activities [56].

Vaccination with rNDV-H5 has been previously demonstrated to offer enhanced protection against AI-infection to chickens carrying NDV-specific maternal antibodies, suggesting that viral entry into the host cell may have been partially H5-dependent [20]. In addition to these findings, the structural differences observed between rNDV-H5 and native NDV LaSota raised the question of their impact on the immune sensing of the recombinant virus by the host immune system. Elucidating the activation of PRRs by rNDV-H5 is fundamental for improving our understanding of protection outcomes previously demonstrated with these vaccines. The results of the present study demonstrated that the innate immune sensing of rNDV-H5 is mediated by TLR3, MDA5, and LGP2 receptors during early infection of CEFs and TOCs, resulting in the induction of IFNβ expression in CEFs. The activation and the antiviral effect of TLR3 during NDV LaSota infection have been previously observed in chicken embryo fibroblast cell line overexpressing TLR3 [57]. The infection with rNDV-H5 induced a dramatic increase in TLR3 expression in comparison to NDV LaSota, suggesting that the expression of exogenous H5 by the vector modified the way it is recognized by the innate immunity sensors. This raised the hypothesis that reduced F expression and the presence of H5 might impact the entry pathway by partly promoting H5-dependent endocytosis during the early phase of infection, instead of the direct fusion entry predominantly used by NDV. Nevertheless, rNDV-H5 retains the ability to induce cytoplasmic PRRs MDA5 and LGP2, demonstrating the presence of genetic material in the cytoplasm. The expression of TLR7, which also senses viral RNA within endosomal compartments, has been previously detected in B cell-like DT40 and macrophage-like HD11 cell lines but not in CEFs [58]. Moreover, the expression of TLR7 previously examined in NDV LaSota-infected HD11 was not found to be altered [59]. The latter finding is in accordance with the lack of detection of TLR7 activation observed in the present study in CEFs and TOCs after NDV LaSota or rNDV-H5 infection. Another hypothesis that may partially explain the differences in innate immunity-related gene expression observed between NDV and rNDV-H5 infections is the potential presence of defective interfering particles (DIPs). Indeed, the generation of DIPs during NDV genome amplification has been previously demonstrated [60]. These particles are potent activators of innate immunity [61,62]. It could be considered that the H5 insertion destabilized the rNDV-H5 genome and induced the generation of immunostimulatory DIPs. Finally, Paramyxoviruses use a mechanism involving the V protein to evade the host’s innate immune responses. The V protein is produced by RNA editing of the phosphoprotein (P) gene [63]. The V protein antagonizes the induction of type-I IFNs by binding both MDA5 and LGP2 [64,65,66], which was demonstrated to be correlated with NDV virulence [67]. V protein expression could be investigated to evaluate whether the insertion of H5 downstream of P may have influenced the expression of V by rNDV-H5 and contributed to the increased expression of IFNβ during cell infection by rNDV-H5.

## 5. Conclusions

Overall, this study demonstrated that the expression of a recombinant H5 by a recombinant NDV can not only impact its biological and structural characteristics but also induce changes in the recognition of the vector by innate immunity. These results confirm the need to systematically investigate the impact of the expression of a foreign gene by commonly used vaccine vectors, such as NDV, in order to improve the understanding of the protection against strains associated with the foreign gene and the vector itself.

## Figures and Tables

**Figure 1 vaccines-09-00758-f001:**
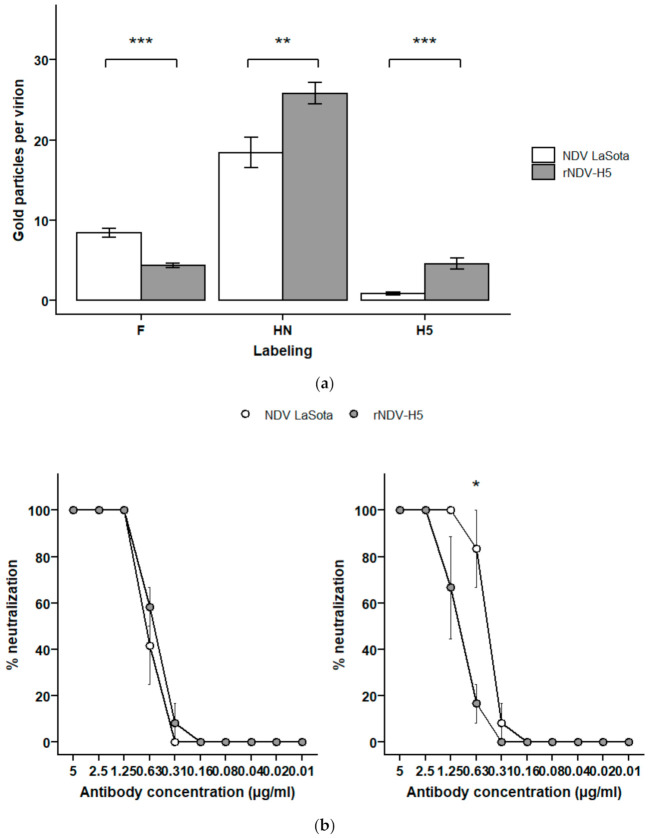
Expression of surface glycoproteins by NDV LaSota and rNDV-H5. (**a**) Proportion of NDV LaSota and rNDV-H5 particles expressing F, HN, and H5. Immunogold labeling of viral particles was performed with anti-HN, anti-F (1C3, IgG1), and anti-AIV H5 (5A1, IgG1) mAbs. The number of gold particles was counted and normalized per virion surface unit of 55,000 nm^2^. Results are expressed as the mean ± standard error of the mean (SEM). (**b**) Comparison of NDV LaSota and rNDV-H5 in a neutralization test using anti-F (left panel) and anti-HN (right panel) monoclonal antibodies. Monoclonal antibodies were two-fold serially diluted and incubated with CEFs for 24 h at 37 °C. Percentages of neutralization are expressed for NDV LaSota (white) and rNDV-H5 (grey) according to the antibody dilutions. * *p* < 0.05, ** *p* < 0.01, *** *p* < 0.001.

**Figure 2 vaccines-09-00758-f002:**
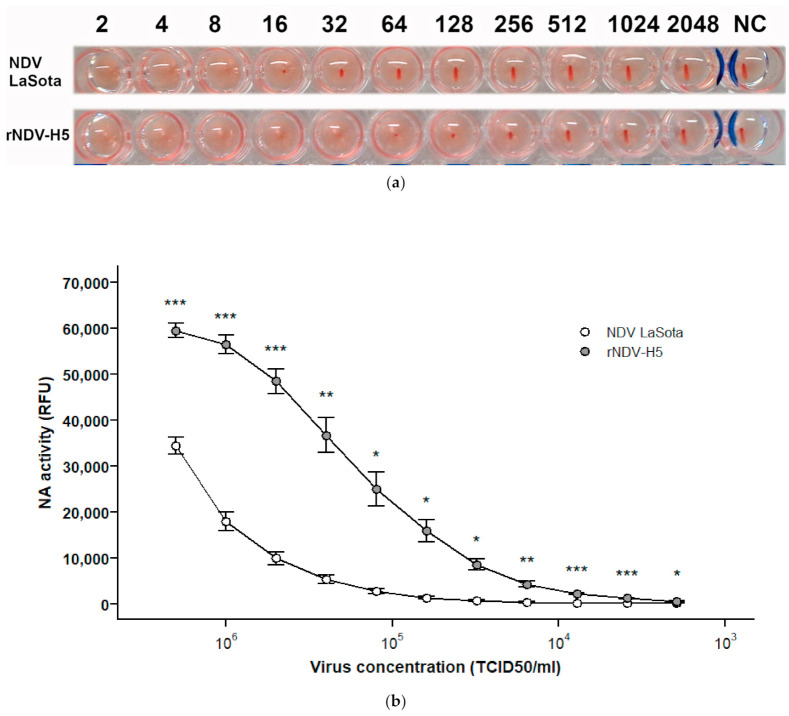
Hemagglutinating and neuraminidase activities of NDV LaSota and rNDV-H5. (**a**) Hemagglutination of chicken erythrocytes by two-fold serially diluted NDV LaSota (107 TCID50/mL) and rNDV-H5 (107 TCID50/mL). NC, negative control. (**b**) Neuraminidase activity of two-fold serially diluted NDV LaSota and rNDV-H5 viruses starting at a titer of 10^7^ TCID50/mL.* *p* < 0.05, ** *p* < 0.01, *** *p* < 0.001.

**Figure 3 vaccines-09-00758-f003:**
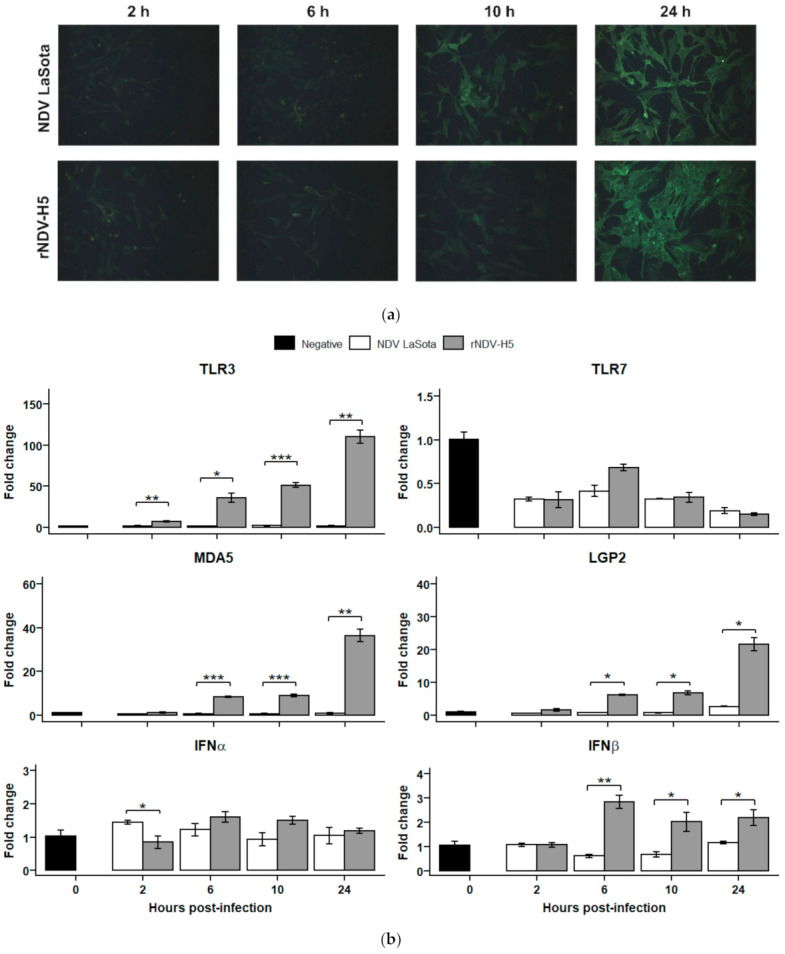
PRRs and type-I IFNs expression in NDV LaSota- and rNDV-H5-infected CEFs. (**a**) CEFs were infected with NDV LaSota and rNDV-H5 (MOI = 1) and observed by fluorescence microscopy using an anti-F antibody (IC3, IgG1) at 2, 6, 10, and 24 hpi. (**b**) Relative expression of TLR3, TLR7, MDA5, LGP2, IFNα, and IFNβ was determined in CEFs infected with NDV LaSota and rNDV-H5 (MOI = 1) at 2, 6, 10, and 24 hpi. The data were normalized to HMBS and RPLP0 expression, calculated according to the 2^-ΔΔCT^ method, and presented ± standard error of the mean. * *p* < 0.05, ** *p* < 0.01, *** *p* < 0.001.

**Figure 4 vaccines-09-00758-f004:**
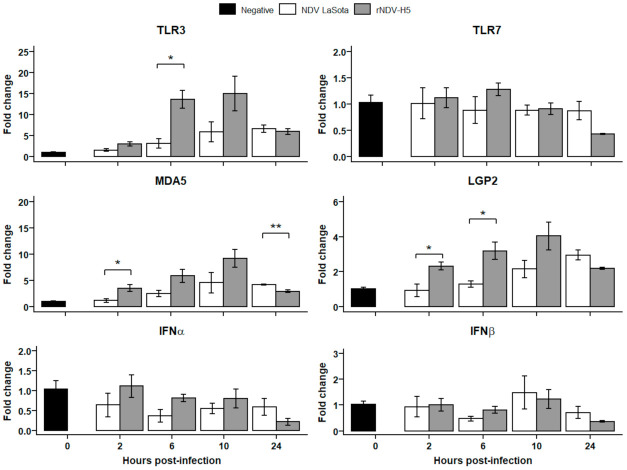
PRRs and type-I IFNs expression in NDV LaSota- and rNDV-H5-infected TOCs. Relative expression of TLR3, TLR7, MDA5, LGP2, IFNα, and IFNβ was determined in TOCs infected with NDV LaSota and rNDV-H5 (MOI = 1) at 2, 6, 10, and 24 hpi. The data were normalized to HMBS and RPLP0 expression, calculated according to the 2^−ΔΔCT^ method, and presented ± standard error of the mean. * *p* < 0.05, ** *p* < 0.01.

## Data Availability

The data presented in this study are available on request from the corresponding author.

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
