# Peer review of "The Expression of Hemagglutinin by a Recombinant Newcastle Disease Virus Causes Structural Changes and Alters Innate Immune Sensing"

_vaccines, 2021, doi:10.3390/vaccines9070758_

Round 1
Reviewer 1 Report
68-70: Poor sentence construction. Revise.
85-86: Add reference for the role of cytokines in the development of the adaptive immune responses.
Overall, the manuscript is well written and it addresses a crucial question of foreign antigen presentation on the surface of NDV. However, the discussion does not really address the possible reason why the recombinant virus induced higher levels of innate immune gene expression compared to the wildtype virus. The speculation about V protein control of innate immune responses is ok but it is also possible for the recombinant virus to accumulate large quantities of defective-interfering particles (due to distabilazation of the genome) that may enhance induction of innate immune responses.
Author Response
Dear Reviewer,
The authors thank you for the helpful remarks and comments you provided. We believe that the manuscript has been improved significantly as a result of the revision.
Please find below the answers to your comments.
On behalf of all the authors,
Yours sincerely,
Fiona Ingrao
Comments and Suggestions for Authors
68-70: Poor sentence construction. Revise.
>>> Changes have been made in the revised version of the manuscript (lines 70-73).
85-86: Add reference for the role of cytokines in the development of the adaptive immune responses.
>>> Changes have been made in the revised version of the manuscript (lines 70-73).
Overall, the manuscript is well written and it addresses a crucial question of foreign antigen presentation on the surface of NDV. However, the discussion does not really address the possible reason why the recombinant virus induced higher levels of innate immune gene expression compared to the wildtype virus. The speculation about V protein control of innate immune responses is ok but it is also possible for the recombinant virus to accumulate large quantities of defective-interfering particles (due to distabilazation of the genome) that may enhance induction of innate immune responses.
>>> This is a very interesting remark and the authors agree this point should be discussed. A discussion of this point has been added to the revised manuscript (lines 380-385).
Reviewer 2 Report
The manuscript “The Expression of Hemagglutinin by a Recombinant Newcastle Disease Virus Causes Structural Changes and Alters Innate Immune Sensing” by Ingrao and co-authors compares the structural and functional characteristics as well as the immune responses of rNDV-H5 vaccine and parental NDV LaSota strain. Differences have been noted in term of the F and HN particles released and gene(s) expression. The manuscript is well written and presented. Few points should be considered and addressed by the authors prior to being submitted for publication as detailed below:
General points:
-In the introduction and M&M: the authors used an old HPAIV H5N1 from clade 1 from Vietnam. It is important to describe in the introduction the significance of this clade and why did the authors select this virus? Is it still circulating? In which countries? Why did the author use this strain “A/Crested-eagle/Belgium/01/2004” not the same Vietnamese strain?
-How do the author relate the HA/NA activity to the only expressed HA from the rNDV-H5? This part should be discussed in more details.
Minor points:
- Line 2: “reecombinant” should read as “recombinant”.
-Line 36: Please add reference!
-Line 40: “have drawbacks that have been previously reviewed”. For example?
-Line 48: “promising viral vector vaccine candidate against human diseases”. Please give example?
-Line 58: “As a paramyxovirus, NDV”. update to the new taxonomy “avulavirus”.
-Line 153, 163, 169: should read with the company name, followed by the country in all.
-Line 182: “ThermoFisherScientific” read as “Thermo Fisher Scientific” and add the country to be consistent with line 177 “(Applied Biosystems, Lennik, Belgium)”
-Line 211: “compare” to “compared”
-Line 226: Figure 1(a) define the x-axis.
Line 269: the contrast of figure 3(a) should be improved.
-Line 330: “Nayak et al”. Year is missing.
-Line 343: “avian influenza” to “AI”
Author Response
Dear Reviewer,
The authors thank you for the helpful remarks and comments you provided. We believe that the manuscript has been improved significantly as a result of the revision.
Please find below the answers to your comments.
On behalf of all the authors,
Yours sincerely,
Fiona Ingrao
Comments and Suggestions for Authors
The manuscript “The Expression of Hemagglutinin by a Recombinant Newcastle Disease Virus Causes Structural Changes and Alters Innate Immune Sensing” by Ingrao and co-authors compares the structural and functional characteristics as well as the immune responses of rNDV-H5 vaccine and parental NDV LaSota strain. Differences have been noted in term of the F and HN particles released and gene(s) expression. The manuscript is well written and presented. Few points should be considered and addressed by the authors prior to being submitted for publication as detailed below:
General points:
-In the introduction and M&M: the authors used an old HPAIV H5N1 from clade 1 from Vietnam. It is important to describe in the introduction the significance of this clade and why did the authors select this virus? Is it still circulating? In which countries? Why did the author use this strain “A/Crested-eagle/Belgium/01/2004” not the same Vietnamese strain?
>>> The detection of H5 by the 5A1 antibody had been previously validated using the A/Crested-eagle/Belgium/01/2004 strain. However, as these internal controls were not included in the manuscript, we prefer to delete the sentence mentioning this strain in the "Materials and Methods" section (lines 108-109).
The clade 1 A/Vietnam/1203/04 strain, whose H5 is inserted into rNDV-H5, has a high zoonotic potential (Steven et al., 2006, DOI: 10.1126/science.1124513) but has not been actively circulating since 2013-2014 (Suttie et al., 2019, DOI: 10.1371/journal.pone.0226108; Smith et al., 2015, DOI: 10.1111/irv.12324; Horm et al., 2016, DOI: 10.1038/emi.2016.69). However, our study aimed to characterize the impact of an H5 insertion on the recombinant NDV vector, independently of the AIV clade. Since an influence of the HPH5 clade type is not expected, we did not feel it required to mention more details about the Vietnamese strain in the "Introduction" section.
-How do the author relate the HA/NA activity to the only expressed HA from the rNDV-H5? This part should be discussed in more details.
>>> The neuraminidase activity of NDV is carried by the HN protein. We apologise for the lack of clarity of this information in the manuscript. To improve the reader's understanding, we have replaced "NA activity" with "neuraminidase activity" throughout the manuscript.
Minor points:
- Line 2: “reecombinant” should read as “recombinant”.
>>> Changes have been made in the revised version of the manuscript (line 2).
-Line 36: Please add reference!
>>> Changes have been made in the revised version of the manuscript (line 36).
-Line 40: “have drawbacks that have been previously reviewed”. For example?
>>> Changes have been made in the revised version of the manuscript (line 41).
-Line 48: “promising viral vector vaccine candidate against human diseases”. Please give example?
>>> Changes have been made in the revised version of the manuscript (lines 49-50).
-Line 58: “As a paramyxovirus, NDV”. update to the new taxonomy “avulavirus”.
>>> Changes have been made in the revised version of the manuscript (lines 60-61).
-Line 153, 163, 169: should read with the company name, followed by the country in all.
>>> Changes have been made in the revised version of the manuscript (lines 119-120, 171, 172).
-Line 182: “ThermoFisherScientific” read as “Thermo Fisher Scientific” and add the country to be consistent with line 177 “(Applied Biosystems, Lennik, Belgium)”
>>> Changes have been made in the revised version of the manuscript (line 185).
-Line 211: “compare” to “compared”
>>> Changes have been made in the revised version of the manuscript (line 215).
-Line 226: Figure 1(a) define the x-axis.
>>> Changes have been made in the revised version of the manuscript (line 230).
Line 269: the contrast of figure 3(a) should be improved.
>>> Changes have been made in the revised version of the manuscript (line 275).
-Line 330: “Nayak et al”. Year is missing.
>>> Changes have been made in the revised version of the manuscript (line 329).
-Line 343: “avian influenza” to “AI”
>>> Changes have been made in the revised version of the manuscript (line 353).